# Biological Pathway-Derived TMB Robustly Predicts the Outcome of Immune Checkpoint Blockade Therapy

**DOI:** 10.3390/cells11182802

**Published:** 2022-09-08

**Authors:** Ya-Ru Miao, Chun-Jie Liu, Hui Hu, Mei Yang, An-Yuan Guo

**Affiliations:** 1Center for Artificial Intelligence Biology, College of Life Science and Technology, Huazhong University of Science and Technology, Wuhan 430074, China; 2Research Institute of Huazhong University of Science and Technology in Shenzhen, Shenzhen 518000, China

**Keywords:** immunotherapy, TMB, SERPINB3

## Abstract

Although immune checkpoint blockade (ICB) therapies have achieved great progress, the patient response varies among cancers. In this study, we analyzed the potential genomic indicators contributing to ICB therapy response. The results showed that high tumor mutation burden (TMB) failed to predict response in anti-PD1 treated melanoma. SERPINB3 was the most significant response-related gene in melanoma and mutations in either SERPINB3 or PEG3 can serve as an independent risk factor in melanoma. Some recurrent mutations in CSMD3 were only in responders or non-responders, indicating their diverse impacts on patient response. Enrichment scores (ES) of gene mutations in 12 biological pathways were significantly higher in responders or non-responders. Next, the P-TMB calculated from genes in these pathways was significantly related to patient response with prediction AUC 0.74–0.82 in all collected datasets. In conclusion, our work provides new insights into the application of TMB in predicting patient response, which will benefit to immunotherapy research.

## 1. Introduction

Cancer immunotherapy with immune checkpoint blockade (ICB) targeting programmed cell death protein 1 (PD-1) or cytotoxic T lymphocyte-associated antigen 4 (CTLA-4) has revolutionized cancer treatment in different malignancies [1]. Despite the remarkable success of ICB therapy, only a small number of patients respond to ICB therapy and the response rates vary across tumor types with 31–44% in advanced melanoma, 22–25% in renal cell carcinoma, and 19–20% in non-small-cell lung cancer (NSCLC) patients [2,3]. Besides, predictive biomarkers for patient response to ICB therapy remain unclear. Past efforts in exploring the differences between responders and non-responders have accumulated some sequencing data at genomic and transcriptional levels [4,5]. Based on these data, some single genomic biomarkers and gene expression models were constructed for response prediction of ICB therapy.

On the transcriptional level, the expression of PD-L1 and IFN-γ were reported to be associated with ICB response [6,7], which were limited in NSCLC, melanoma, or renal-cell cancer. The predictive power needs to be determined by further studies across different tumor types [6,7]. In addition to single biomarkers, gene expression models were constructed for ICB therapy response prediction. TIDE predicts the outcome of melanoma and NSCLC patients treated with anti-PD1 or anti-CTLA4 by signatures of T cell dysfunction [8]. IMPRES can predict the ICB response of melanoma patients by their immune interactions [9]. ICGe predicts patient response by the expression of immune checkpoint genes [10]. ImmuCellAI predicts response by the abundance of immune cell subtypes [11]. However, these models have often been validated in small clinical cohorts and the stability of gene expression models needs to be validated in larger data sets.

On the genome level, tumor mutation burden (TMB), which is measured by the number of nonsynonymous mutations per megabase (Mb) of the genome, was reported as an independent predictor of response to immunotherapy in diverse cancers [12,13,14]. Recent studies have focused on the relevance of tumor neoantigens in ICB therapy [12,15]. Besides, HLA type and tumor aneuploidy may have additional predictive value for the response or resistance to ICB therapies [16,17]. However, there was no certain threshold of TMB between ICB therapy responders and non-responders. The TMB threshold of 10 muts/Mb, which was reported as a reasonable predictor of response in NSCLC treated with nivolumab plus ipilimumab [18], may not apply to all immune checkpoint inhibitors or cancer types [13,19]. In addition, it was reported that selective oncogenic signaling pathway alterations such as β-catenin signaling pathway activation or PTEN loss were associated with diminished T-cell infiltration, thus affecting antitumor response [20,21]. More recently, David et al. revealed that DNA repair gene mutations can serve as predictors of ICB response beyond TMB [22]. Furthermore, evidence from ICB-treated patients with NSCLC and melanoma suggested that only clonal neoantigens were associated with clinical benefit [22,23].

Given the limitations of existing biomarkers, there is an urgent need to develop robust response-related indicators applicable to multiple datasets and cancer types. A lot of work illustrated the importance of genomic factors in ICB therapy. However, rare response prediction model was built based on genomic data. In this study, we collected genomic data as well as the clinical information of ICB-treated patients with melanoma and NSCLC from six published studies [12,13,14,24,25,26]. By performing genomic analysis, we identified and evaluated factors correlated with patient response or survival. We also introduced a new scoring system P-TMB, which was calculated based on mutations in genes from selected biological pathways and was more robust than TMB in predicting patient response.

## 2. Materials and Methods

### 2.1. Raw Sequencing Data and Clinical Information Collection

Raw Whole-Exome Sequencing (WES) data and clinical information of melanoma and NSCLC samples treated with anti-PD-1, anti-CTLA-4, or a combination of anti-PD1 and anti-CTLA4 were collected from NCBI, dbGAP, or EBI using the accession IDs reported in published papers (Appendix A). Clinical response information of patients in cohorts mentioned above was under criteria of Response Evaluation Criteria In Solid Tumors (RECIST), RECIST 1.1 and irRECIST. In this study, responders (R) were composed of complete response (CR), partial response (PR), and patients with long-term-benefit; non-responders (NR) were patients with stable disease (SD), progressive disease (PD), or minimal-or-no benefit. For the analyses presented in this study, out of the 287 patients with complete clinical data, 211 patients had advanced melanoma (stage IV) and 76 patients had advanced non-small cell lung cancer (NSCLC) (stage IV). For melanoma, the ages of the patients ranged from 18 to 90 (average: 60 ± 14), and male patients accounted for 67.7%. Majority tumor samples were taken from melanoma metastasis (90%), and the left were taken from the primary melanoma. All samples were taken at stage IV. For NSCLC, age of the patients ranged from 41 to 85 (average: 63 ± 9), and male patients accounted for 51.3%. All samples were taken from the lung tumor at stage IV. Furthermore, the clinical endpoint used in these analyses was overall survival, defined as the length of time from treatment start to time to event. All clinical data were obtained from the original studies. Additional details regarding these patients can be found in the original publications [12,13,14,24,25,26].

### 2.2. Somatic Mutation Calling and Annotation

The tumor-normal paired samples were extracted for somatic mutation detection. Raw sequence reads were transformed to bam format file by using tools packed in GATK4 (Genome Analysis Toolkit, v4) [27]. In GATK4, reads were assembled over several different k-mer sizes; for reads in low-complexity regions, large k-mers will be used, which may fix the mapping issues in low-complexity regions to some extent. The whole mutation calling process includes the following steps: (1) Raw sequence reads were mapped to reference human genome GRCh38; (2) the “MarkDuplicates” step was performed to mitigate biases introduced by data generation steps; (3) Recalibrate the base quality scores to improve the accuracy of the variant calling step. Next, MuTect2 [28] was used to generate somatic mutations by comparing BAM files from tumor and matched normal samples using default parameters, the possible germline variants were filtered with the parameter “--germline-resource” by using the gnomAD dataset “af-only-gnomad.hg38.vcf.gz” as a reference. The dbSNP was used to filter the SNP with “Homo_sapiens_assembly38.dbsnp138.vcf” as reference. Finally, all somatic mutations were annotated by using Ensembl variant effect predictor (VEP) [29].

### 2.3. HLA Class Typing and Neoantigen Prediction

For each patient, OptiType [30] (v1.3.1) was used to estimate the HLA-I major loci (A, B, and C) at four-digit resolution. All possible 8–11 mer peptide sequences resulting from somatic nonsynonymous mutations in the protein coding region were extracted. NetMHCpan (v4.0) [31] was used to compute the binding affinities of HLA and neoantigen peptides. Antigens with HLA binding affinity of <50 nM were considered as strong binding neoantigens and those with 50–200 nM as weak binding neoantigens.

### 2.4. Response-Associated Gene Selection

Genes significantly correlated with patient response were selected based on the following conditions: (1) *p*-value of Fisher’s exact test between gene mutation in responders and non-responders is less than 0.05; (2) The fold change between mutation frequency in responders and non-responders is greater than 1.5.

### 2.5. TMB and P-TMB Calculation

TMB of samples were calculated by the number of nonsynonymous somatic mutations (including “missense mutation”, “inframe insertion”, “inframe deletion”, “frameshift insertion”, “frameshift deletion”, “start lost”, “stop gained”, “stop lost”, “splice_acceptor_variant”, and “splice_donor_variant”) per megabase in coding regions. P-TMB was derived from gene mutations in selected biological pathways. First, a total of 209 biological pathways including 186 KEGG pathways, 17 immune pathways from ImmPort [32], and 6 DNA repair pathways [22] were collected. Normalized enrichment scores (NES) of mutated genes in each pathway per sample were calculated by GSVA (Gene Set Variation Analysis) function [33] in each dataset as follows. First, the mutation frequency matrix (M) of each dataset was produced by calculating the mutation frequency of genes in samples. Thus, M is an m × n matrix with m representing the number of samples and n representing the number of mutated genes in all samples. The number in matrix M is the mutation frequency of gene in each sample. By taking the M and gene list of 209 pathways as the input of GSVA, we can obtain the NES of gene mutations in each pathway per sample. Following this, the Mann–Whitney U test was performed between the ES of responders and non-responders. Next, pathways significantly enriched in responders in more than three datasets were denoted as positive pathways (PP, Mann–Whitney U test *p*-value < 0.05). Oppositely, pathways significantly enriched in non-responders in more than three datasets were denoted as negative pathways (NP).

After identifying the PP and NP pathways in samples, we considered if it was possible to use the TMB calculated from genes in PP and NP pathways to predict the patient response. Thus, we developed the “Pathway-derived Tumor Mutation Burden (P-TMB)” as follows:$$P\text{-}\mathrm{TMB}=(\frac{\sum_{i=1}^{n}P}{\sum_{i=1}^{n}LP}-\frac{\sum_{j=1}^{m}N}{\sum_{j=1}^{m}LN})\times 10^{6}$$
where *P* and *N* represent the number of nonsynonymous mutations of each gene in PP and NP, respectively; *n* and *m* are the number of genes in PP and NP, respectively. *LP* and *LN* represent the length of each gene in PP and NP, respectively.

### 2.6. P-TMB Threshold Calculation

Youden Index–associated P-TMB cutoffs in each dataset were calculated using the “cutpointr” package [34]. Then, the best threshold of P-TMB in all datasets was the average value of cutoffs in all datasets.

### 2.7. TCGA Cancer Samples Collection

Mutation data and clinical information of TCGA cancer samples were downloaded from “firehose” of Broad Institute (http://gdac.broadinstitute.org/, accessed on 6 November 2019).

### 2.8. Statistical Analysis

Significant tests of binary molecular features such as wild-type or mutant gene with patient response were carried out using Fisher’s exact test. Differences in continuous variables between responders and non-responders were assessed using a non-parametric Mann–Whitney U test. Correlation analysis was performed with the “psych” package [35]. ROC analyses were performed using the “pROC” package [36]. Survival and cox regression analysis were performed with the “surminer” package. All statistical analyses were carried out in R.

## 3. Results

### 3.1. The Prediction Ability of Tumor Mutation Burden in Patient Response Is Limited

Whole genome or exome sequencing (WGS/WES) data and clinical response information were available for 287 melanoma or NSCLC samples treated with the drug anti-PD1, anti-CTLA4, or their combination (Figure 1A, Appendix A). Following this, the relationships between patient response and TMB or neoantigen, which were produced by tumor mutation, were investigated. Among all six collected datasets, we observed that TMB was significantly correlated with patient response in three of them (Figure 1B,C). In addition, the number of total neoantigens and neoantigens from missense mutations can predict patient response in melanoma CTLA4 and NSCLC PD1 cohorts (Figure 1B). Interestingly, for melanoma patients treated with anti-PD1, although there was no significant difference in TMB between responders and non-responders, the count of neoantigens from in-frame insertion mutation was significantly higher in responders (*p*-value 0.021, Figure 1D), which may be a response signature in anti-PD1 treated melanoma. Although the association between TMB and response to ICB has been demonstrated by some studies, we observed that TMB failed to predict patient response in some cohorts, such as anti-PD1 treated melanoma. Furthermore, the TMB of most patients was less than 10 per MB (Figure 1E) and for samples from different cohorts with the same cancer type and ICB therapy, the TMB calculated using the uniform rule was varied (Figure 1F). In conclusion, the prediction power of TMB to response varies by cancer type or therapy, which is also reported by McGrail et al. and there was no certain threshold of TMB between responders and non-responders as their TMB distributions overlapped substantially [37]. Thus, further studies need to be carried out to investigate the correlation between mutation and response.

### 3.2. SERPINB3 Is the Most Significant Response-Related Gene in Melanoma

Following these steps, we focused on the impact of single gene mutation on patient response. Previous studies indicated that patients with KRAS mutation had remarkable clinical response to PD-1 inhibition, especially those with accompanying TP53 mutation [38]. In this work, we systematically analyzed the impact of gene mutations on patient response in melanoma and NSCLC, respectively. Ultimately, genes significantly correlated with patient response were selected based on the difference of mutation frequency between responders and non-responders (detailed in the method section). The selected response-related genes in melanoma and NSCLC were presented in Figure 2A and 2B, respectively. The results showed that SEPRINB3 was the most significant response-related gene in melanoma, with a fold change of mutation frequency greater than 5 between R and NR (Figure 2A). Responders with SERPINB3 mutation accounted for 7.6% of all samples, while for non-responders, this number was 2.4% (Figure 2C). Furthermore, Riaz et al. reported that patients with recurrent SERPINB3 and SERPINB4 mutations respond to anti-CTLA4 immunotherapy [39], which demonstrates the potential of SERPINB3 to serve as a response signature in melanoma. Moreover, among those response-related genes, mutations of PAPPA2, LRP1B, KALRN, TTN, and MUC16 were reported to be associated with higher TMB and patient prognosis (genes with green color in Figure 2A,B) [40,41,42,43]. Thus, we further investigated the association of mutation of these genes with TMB, the result indicated that mutations of MUC16, TTN, PAPPA2, LRP1B, and KALRN were significantly associated with higher TMB in all six independent cohorts, which were consistent with published studies (Appendix A, Mann–Whitney U test *p*-value < 0.05) [40,41,42,43].

For NSCLC, the most significant response-related genes included DCHS2 and COL11A2 (Figure 2B). The frameshift mutation of DCHS2 was reported to be associated with high microsatellite instability in gastric and colorectal cancer [44]. Among all NSCLC samples, the DCHS2 mutated responders accounted for 12%, and non-responders with DCHS2 mutation only accounted for 1% of all samples (Figure 2C). Furthermore, the reported TMB-related genes including PAPPA2, KALRN, and TTN were significantly correlated with patient response in NSCLC, which were also identified in melanoma. In addition, we observed some other shared response-related genes between melanoma and NSCLC including RP1, FLNC, MYT1L, ANK1, BSN, COL1A2, PCDHA10, and KCNH5 (genes with red color in Figure 2A,B), which have the potential to predict patient survival in multiple cancer types. Furthermore, mutation types of the top 5 most significant response-related genes in melanoma and NSCLC include missense mutation, stop gained mutation and frameshift deletion, among which the missense mutation accounted for the largest proportion (Figure 2D,E). Moreover, HLA and B2M genes play key roles in the antigen presentation process and influence patient response to ICB therapy [16,45], thus, we investigated the impact of HLA and B2M genes on patient response. The results indicated that HLA-DRB5 had significantly higher mutation frequency in responders and the B2M mutations only occurred in non-responders in melanoma (Figure 2F).

Interestingly, 777T/I, 519L/F, and 1312G/R mutations on PAPPA2, and 231P/L and 171E/K mutations on CSMD3 were only observed in responders with melanoma or NSCLC (Appendix A). On the contrary, 182E/K and 1684P/T mutations on PAPPA2 and 645E/K mutation on CSMD3 were only observed in non-responders, which illustrated that mutations derived from the same gene may have diverse impacts on patient response, suggesting that mutation quality is more important than mutation quantity. Considering that the mutation frequency of genes that serve as prediction signatures should not be too low, we investigated the mutation frequency of SERPINB3, DCHS2, and the above melanoma and NSCLC shared response-related genes in TCGA melanoma (SKCM) and lung cancer data. The result indicated that the frequency of most genes was larger than 10% in SKCM and 5% in TCGA lung cancers (Figure 2G).

### 3.3. Distribution of HLA Subtypes and Neoantigens in ICB Treated Samples

Besides gene mutation, HLA subtype and neoantigens were reported to impact on the response of ICB therapy [16,46]. In this work, we observed some recurrent HLA subtypes and neoantigens in samples from different datasets (Appendix A). The result showed HLA-A*02:01 is the most frequent HLA subtype in four datasets, followed by HLA-A*03:01 and HLA-B*44:02 (Appendix A). In addition, we observed that HLA-A*01:01, HLA-A*03:01 and HLA-B*35:01 were significantly correlated with patient survival (log-rank test *p*-value < 0.05), HLA-A*01:01 and HLA-B*08:01 were significantly correlated with patient response (fisher’s exact test *p*-value < 0.05). For neoantigen peptides, the “KIGDFGLATEJ” peptide produced by BRAF V600E mutation was frequently observed in four melanoma datasets with frequency larger than 0.05 in each dataset (Appendix A).

### 3.4. Mutations in Either SERPINB3 or PEG3 Can Serve as Independent Risk Factor in Melanoma

Gene mutations related to good survival may also contribute to patient response during the therapy. However, no single gene was observed significantly correlated with patient survival in all cohorts. Following, we investigated the impact of gene pairs (represented as “A|B”, which means the mutation of gene A or gene B) on patient survival. By performing survival analysis based on gene pair mutation, we observed that the mutation of nine gene pairs, such as “SERPINB3|RELN”, “KAT6B|RELN”, and “SERPINB3|PEG3”, were significantly correlated with patient survival in all four independent melanoma datasets (Figure 3A, log-rank test *p*-value < 0.05). Moreover, the combination of these genes also had relatively high mutation frequency (median frequency higher than 0.1) in all melanoma cohorts (Figure 3B). Following multivariate Cox analysis using age, gender and the mutation of the nine gene pairs mentioned above as features, it was shown that the SERPINB3|PEG3 pair was an independent risk factor in melanoma (Figure 3C). SERPINB3 was reported to modulate TGF-beta expression and its high expression indicated poor prognosis in liver cancer [47,48]. PEG3 was also reported as a positive regulator of cancer aggressiveness and angiogenesis [49]. Thus, the mutations of SERPINB3 or PEG3 may lead to longer survival.

Furthermore, we observed that the frequencies of most gene pairs in SKCM and lung cancers from TCGA were larger than 0.2 (Appendix A). Besides, the frequency of TPR|FOXA3 and ABCA 7|DOCK2 were larger than 0.1 in more than 10 cancers (Appendix A). Next, multivariate Cox analysis was performed in TCGA cancers to investigate the impact of gene pairs on survival in other cancers. The result showed KAT6B|RELN pair was an independent risk factor in five cancers (CESC, SARC, LIHC, LUAD, and PRAD) (Figure 3D). The Kaplan–Meier curves showed that the occurrences of ABCA7|DOCK2, CDH6|ARHGEF4, KAT6B|RELN mutations were significantly associated with good prognoses in OV, ESCA, STAD, or LUAD (Figure 3E).

### 3.5. Response Associated Biological Pathways Enriched by Mutant Genes Are Shared between Melanoma and NSCLC

Next, we calculated the NES of gene mutation on biological pathways by GSVA function [33]. Interestingly, the NES of some pathways was significantly higher in responders or non-responders. Pathways with higher NES in responders were denoted as response positive pathways (PPs). Conversely, pathways with higher NES in non-responders were denoted as response negative pathways (NPs). Finally, a total of 12 pathways including 4 PPs (Neutrophil extracellular trap formation, MAPK signaling pathway, Base excision repair, and Mismatch repair) and 8 NPs (Chemokine Receptors, Autoimmune thyroid disease, Regulation of autophagy, Citrate cycle (TCA cycle), DNA checkpoints, TGF-b Family Members Receptors) were observed to have significantly higher NES in responders or non-responders in more than three datasets (Figure 4A,B). Neutrophil extracellular traps were reported to promote tumor growth and metastasis [50], thus its gene mutation may associate with the favorable result of ICB therapy. Furthermore, the mutation in the mismatch repair pathway was reported to indicate durable responses to ICB in urothelial carcinoma and colon cancer [51]. On the contrary, TGF-b Family Members Receptor was attributed to the diminishing of T-cell infiltration, thus impairing the antitumor immune response [52], and its gene mutation may enhance the immune killing process. In addition, the mutation of DNA checkpoints may lead to tumor immune escape, which may cause unfavorable result of treatment. Past efforts have demonstrated that loss of function mutation of JAK-STAT and IFN gamma pathways can lead to the resistance of ICB therapy [53]. However, in this work, there was no significant enrichment of mutated genes in non-responders in these pathways (Appendix A).

### 3.6. The Self-Build P-TMB Was Significantly Associated with Patient Response

As described previously, TMB was not robust in predicting ICB response. Following this, we introduced a P-TMB, which was the TMB calculated from a total of 694 genes in 12 response-associated biological pathways (Figure 4A, Appendix A). As we expected, P-TMB was significantly associated with patient response in all collected melanoma and NSCLC cohorts (Figure 4C, *p*-value < 0.05). P-TMB as a single continuous variable had good predictive value in most cohorts (area under the receiver operating characteristic curve (AUC) ranged from 0.74–0.82, Figure 4D). Most interestingly, a cutoff of 4.3 (average Youden Index-associated cutoffs of six datasets, detailed in method section) can serve as the threshold of ICB response in most cohorts (Figure 4E), the P-TMB of most responders was larger than 4.3 and which was the opposite for non-responders (Figure 4C,E). Furthermore, P-TMB was still significantly correlated with response in an independent melanoma dataset (SRP095809, *p*-value < 0.05, Figure 4F). The AUC of P-TMB in predicting response in SRP095809 was 0.763 (Figure 4F). Moreover, consistent with the previous finding, P-TMB of most responders in SRP095809 were larger than 4.3, which illustrated that P-TMB can serve as a robust predictor of response and the cutoff 4.3 can be the uniform threshold of different cohorts.

## 4. Discussion

Past efforts to analyze the mechanisms or characterize biomarkers of the response to ICB therapy have revealed factors associated with response from genomic-, transcriptional-, epigenetic- and metagenomic levels. In this work, we collected WES/WGS data of tumors or normal tissues from ICB-treated samples of melanoma and NSCLC. By performing genomic analysis, we identified and evaluated factors correlated with patient response or survival. Furthermore, we derived the P-TMB, which was calculated by genes in pathways positive or negative associated with response. Compared with the TMB, P-TMB was more powerful in predicting patient response.

A lot of work has proved that TMB can serve as a biomarker of response to ICB therapy [54,55]. Among the six collected datasets, we observed significantly higher TMB in responders of NSCLC and melanoma samples who received anti-CTLA4 blockade therapy (Figure 1C) but not in anti-PD1 treated melanoma samples. The investigation of Marabelle et al. revealed that a TMB threshold of 10 muts/Mb is a predictor of response in NSCLC [18], which was failed to validate in one of the NSCLC cohort in this study. For melanoma samples from different datasets, the distribution of TMB varied, which indicated that a settled threshold of TMB in response prediction may not be applicable.

In this work, we investigated the impact of single gene mutation and gene mutation in selected biological pathways on patient survival or response, and aimed to represent the TMB using selected genes, which will be more robust in prediction response. From the single gene level, some of PAPPA2 mutations occurred only in responders and other mutations occurred only in non-responders, which demonstrated that even mutations of the same gene may play different roles in the tumor progression and treatment. The result may partially explain the failure of high TMB in predicting patient response. These suggest that not all mutations contribute equally to response and the response scoring system should take more details into consideration. Differently from single genes, some biological pathways were observed to be significantly associated with patient response in multiple datasets. Using genes from selected biological pathways, we developed the P-TMB, which can robustly predict patient response in all melanoma or NSCLC cohorts and the independent validation cohort (Figure 4C–F). Compared with the reported response prediction models based on the transcriptomic data such as TIDE [8] and IMPRESS [9], the P-TMB based on whole-exome data in this work has prediction power in both melanoma and NSCLC samples. In addition, the settled P-TMB threshold of 4.3 has great advantages in the utilization. However, the reported ICB response-related pathways such as JAK-STAT and IFN gamma pathways were not significantly enriched by mutated genes in non-responders collected in this work. It was reported that antitumor immune responses are largely driven by STAT1 and STAT2 induction of type I and II interferons (IFNs). Conversely, STAT3 has been widely linked to cancer cell survival, immunosuppression, and sustained inflammation in the tumor microenvironment [56]. Thus, the function of JAK-STAT pathways is complex. In this work, we also observed that the enrichment score of this pathway showed an opposite trend between the responders and non-responders in different datasets (Appendix A).

While the sample heterogeneity and limited sample size are still major limitations of this work, conclusions obtained from small datasets may not universal. Investigation results indicated that some reported response-related factors failed to validate in this work, implying that larger studies are needed to identify reliable and robust biomarkers of response and intrinsic resistance to ICB therapy. The accuracy of the model is influenced by heterogeneity among patients and the sequencing depth of datasets. Thus, integrating features from different levels using the deep learning-based multi-omics integration method will shed light on the response prediction research, which may improve the robustness of the response prediction model.

## Figures and Tables

**Figure 1 cells-11-02802-f001:**
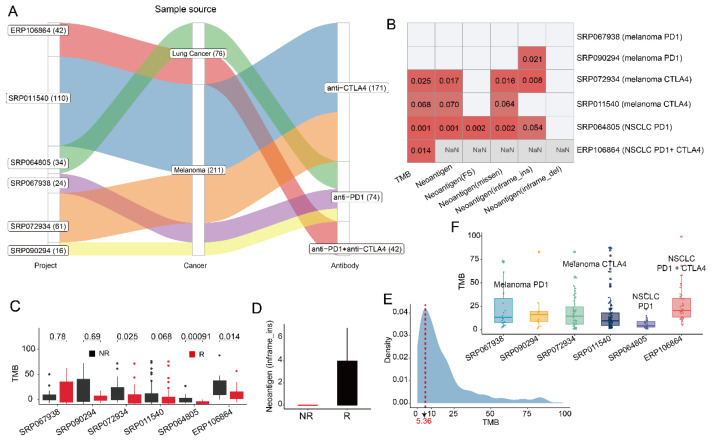
Mutation landscape of samples from different projects. (**A**). Sankey plot of detailed information of data collected in this project. (**B**). Correlation of TMB, neoantigen and neoantigen from different types of mutations with patient response. “NaN” indicates no neoantigen results in NSCLC treated by the combination of anti-PD1 and anti-CLTA4 (the raw sequence data not available). (**C**). Boxplots of TMB between responders and non-responders in different cohorts. TMB was significantly higher in responders of melanoma treated with anti-CTLA4 and NSCLC samples (Mann-Whitney U test *p*-value < 0.05). (**D**). Boxplot of neoantigens from in-frame insertion mutation between responders and non-responders in anti-PD1 treated melanoma samples. (**E**). Density profile of TMB in all samples. (**F**). Boxplots of TMB in samples of two anti-PD1 treated melanoma cohorts, two anti-CTLA4 treated melanoma cohorts, NSCLC treated by anti-PD1, and the combination of anti-PD1 and anti-CTLA4.

**Figure 2 cells-11-02802-f002:**
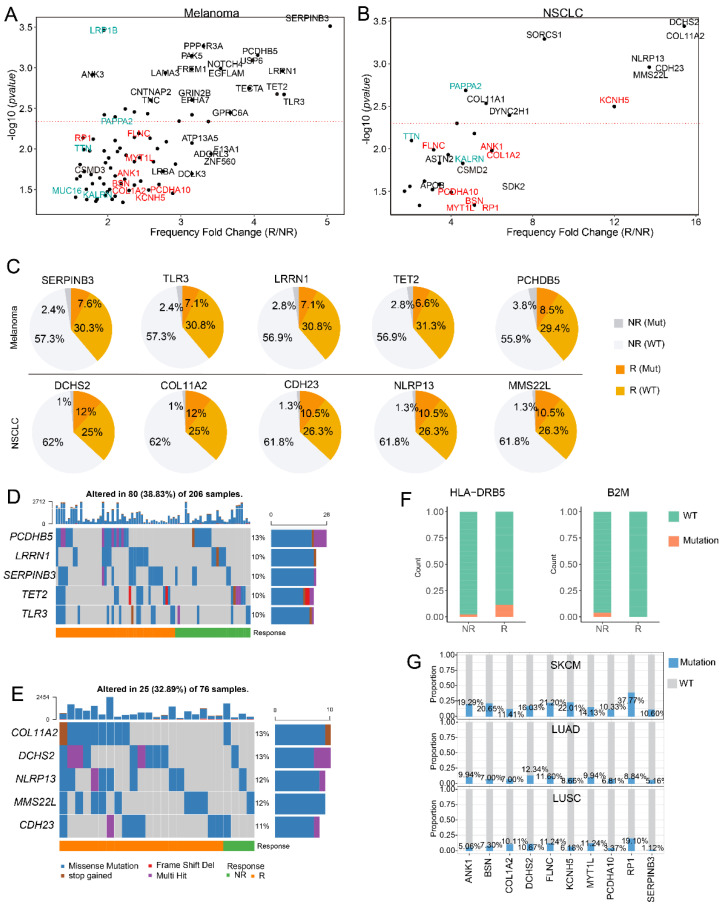
Genes significantly associated with response. (**A**,**B**). Mutation of genes significantly correlated with patient response in melanoma (**A**) and NSCLC (**B**). Genes with red color denote shared genes in melanoma and NSCLC. Genes with green color are TMB-related genes. (**C**). Pie charts of proportions of mutated and wild type samples in response and non-response group for the top 5 most significant response-related genes in melanoma and NSCLC. (**D**,**E**) Oncoplot showing the mutation types of the top 5 most significant response-related genes in melanoma (**D**) and NSCLC (**E**). (**F**). Bar plots showing the mutation frequency of HLA-DRB5 and B2M in responders and non-responders, mutation frequencies of these genes were significantly different between responders and non-responders in melanoma. (**G**). Mutation frequency of response-related genes in TCGA SKCM, LUAD, and LUSC samples.

**Figure 3 cells-11-02802-f003:**
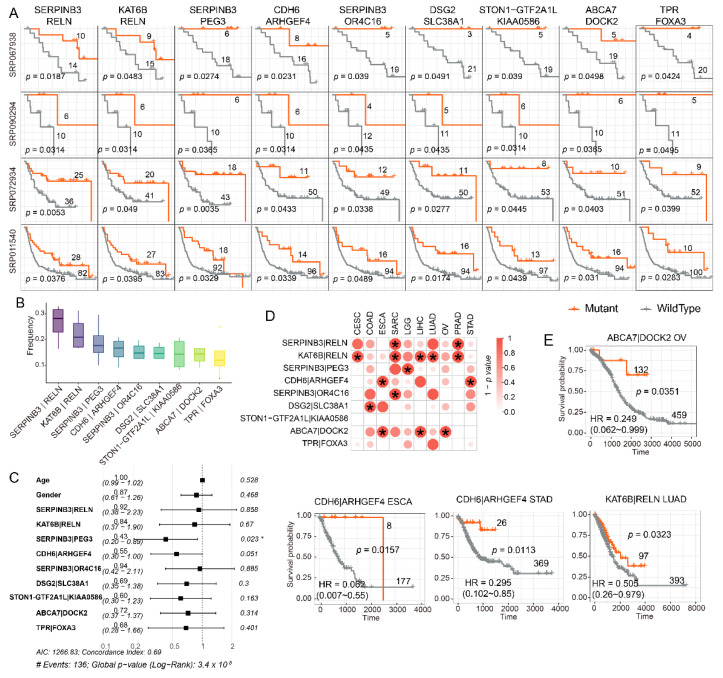
The mutation of gene pairs significantly associated with patient survival. (**A**). Kaplan–Meier curves showing the mutation of 9 pairs of genes associated with longer survival in four independent melanoma datasets. *p*-value was measured by log-rank test. (**B**). Boxplots showing the mutation frequencies of survival-related gene pairs. (**C**). Forest plot showing log hazard ratios (HR) and their 95%-confidence intervals of survival-related gene pairs and clinical indicators (including age and gender) by cox regression analysis. The *p*-value comes from testing the null hypothesis of hazard ratio 1. (**D**). Dot plot showing the effect of survival-related gene pairs on patient survival in TCGA cancer samples. “*” indicates a log rank test *p*-value < 0.05. (**E**). Kaplan–Meier curves showing gene pairs as an independent risk factor in TCGA cancer samples (Likelihood ratio test < 0.05 in cox regression analysis).

**Figure 4 cells-11-02802-f004:**
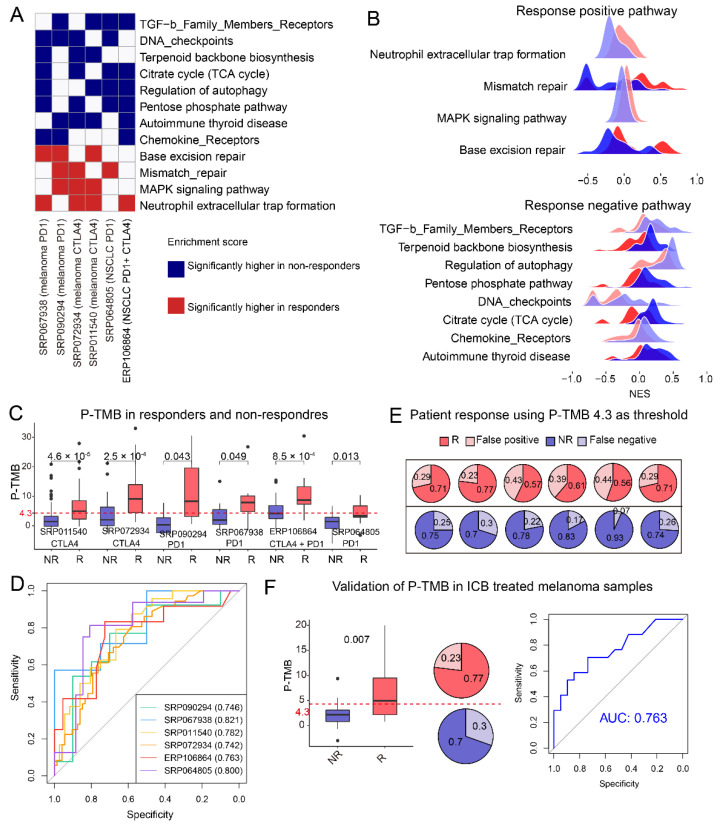
Biological pathways associated with response and pathway-derived TMB. (**A**). Heat map showing biological pathways with a normalized enrichment score (NES) significantly different between responders and non-responders. (**B**). Ridge plots present the NES of mutated genes in response positive pathways and negative pathways in samples from six datasets. NES of mutated genes in non-responders and responders were presented in blue color and red color, respectively. (**C**). Boxplot of distributions of P-TMB in responders and non-responders in all six datasets. (**D**). Receiver operating characteristic (ROC) curves for prediction of patient response using P-TMB as a feature. (**E**). The proportion of responders and non-responders in group with P-TMB > 4.3 (red color) and the opposite (blue color). (**F**). Validation of P-TMB in newly collected ICB-treated melanoma samples (GSE91061).

## Data Availability

Raw sequencing files from previously published cohorts are available at the database of Genotypes and Phenotypes (dbGaP) under the following accession “PRJNA307199”, “PRJNA343789”, “PRJNA306070”, “PRJNA82747”, “PRJNA293912”, “PRJEB24995”.

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
