# Peer review of "Biological Pathway-Derived TMB Robustly Predicts the Outcome of Immune Checkpoint Blockade Therapy"

_cells, 2022, doi:10.3390/cells11182802_

Round 1

Reviewer 1 Report

MAJOR POINTS:

Your study rationale is extremely important. 

Your work and findings in relation to SEPRINB3 is very interesting, in terms of potentially assigning this as a response signature in melanoma

MINOR POINTS

Some language edits e.g.

  • Abstract: 'was failed to predict response' change to 'failed to predict response'
  • Introduction: 'Cancer immunotherapy' not 'The Cancer Immunotherapy'
  • Introduction: Line number 62,63: 'Only a small part of patients respond to ICB' - change to only a small number of patients'. This statement should also be followed by an example of the response rates to make this credible to the reader. ?what are the response rates across tumour types. 
  • The full manuscript should again be thoroughly proofread to ensure no grammatical errors. 

Ensure appropriate referencing through, e.g.

  • Introduction: 'Compared with gene expression from transcriptomic data, gene mutation derived from genomic data is more stable and gene mutation is the leading cause of cancers' - this statement should be backed up with appropriate reference. 
  • Website in line no. 209 should be formally changed to a reference in the bibliography and not included in main text.

Results:

Section 3.3: the term 'prognosis' appears to be used when 'response' may be more correct.

Reviewer 2 Report

In this study Ya-Rumiao et collegues have performed a meta-analysis on genomic data from six published series on patients treated with ICB. 

The analysis refer to the Tumor-mutation burden (TMB) and its relevance to ICB response. The authors conclude that a Pathway-derived Tumor Mutation Burden (P-TMB) scoring system can predict more efficientlly the outcome of ICB treated patients than the traditional TMB approach. The study is of interest, although there are some major concerns, especially regarding the presentation of the results obtained. 

  • Figure 2: What do the "TMB-related genes" stand for (the ones in green color)? I understand that all of the genes shown in Fig 2A and 2B refer to genes with different TMB between responders and non-responders. What is the difference in the green-labeled genes?
  • Line 206: This correspnd to 2F and not 2D. 
  • Lines 209-214: This correspnds to 2D instead of 2F. Moreover, in line 213, do the authors mean frequency of MUTATED genes were bigger than 10%?
  • I cannot understand what is the S1 Figure 1 showing? Does it refer to the TMB of response-related genes between responders and non-responders? The title of S1 is misleading. Moreover, the labeling (black/red) in S1 FIG is probably responders vs. non responders instead of WT and mutation? Otherwise the authors should clearly expalin what the fig is depicting.
  • Part 3.3: The authors should clearly explain the criteria used to select/combine the two genes in each gene-pair. Are the genes in each pair functionally related? If not, the approach that a mutation in eiter gene A or gene B is correlating with patient survival  may be arbitrary or random
  • Figure 3: The Figure Legend should be modified to better expalin what is being shown in each part of the Figure. Abbreviations are also missing.
  • The enrichment score (ES) should be described in more detail in the Materials and Methods section. Moreover the survival analysis description is missing from the Statistical analysis part in 2.8.
  • Figure 4: What does the NES stand for? What does the blue and the red color depict?
  • Lines 276-278: TGF-b activation has been correlated with less effective immune responses. Therefore, a higher mutation burden in this family could lead to TGF-b deregualation and stronger immune responses. Can the authors explain why a higher TMB in this family is reported in the non-responders group?
  • Lines 207-209: These findings are interesting and should be further discussed in the discussion section.
  • The authors should carefully check for spelling and grammar errors. 

Reviewer 3 Report

This an interesting study at first glance. However, the study enthusiasm was quickly dampened by the poor analytics and data interpretation: there are many flaws of the manuscript:

Mutation calling: it is unclear whether the authors subtract the results with matched germline controls and whether they filtered the results with SNP databases such as 1000 genomes and dbSNPs. People often removed the low complexity region as well due to the mapping issues.

Figure 2, It is also unclear whether the authors have removed the synonymous mutations, mutations in non-coding region such as UTR and introns, or included all the mutations in the analysis. Authors should also separate the loss of function of mutation such as frameshift/Stop gain and splicing site from missense mutation: in a tumor type with high mutational burden, the majority of the missense mutations are likely bystander/passenger.

Figure 2a, mutation of TET2 and TLR3 enriched in responder are against the current theory of immunotherapy. For example, activation of TLR3 pathway often enhances the response of PD1. Again, authors should display the type of the mutations (missense or stop gain).

Figure 2e, a difference of 2 mutations is meaningless in clinical practice.

Authors should integrate the analysis of HLA/B2M (mutation, copy number and expression) and JAK STAT signaling, as loss of function mutation of these genes will lead to the immune evasion: loss of the HLA/B2M will abolish the TCR binding and T cell killing. In this scenario, mutation burden and neoantigen will not have NO effect on ICB response. Also, authors should fully discuss the key role of JAK STAT and IFN pathways in regulating the ICB response.

Authors identified TTN, MUC16 etc. as a predictive marker: there genes are extremely big, for example, TTN is the largest gene in the human genome with 34350 amino acids. Authors are actually still looking at the mutational burden and the mutation rate of each sample, the reason of enrichment of these genes in the result is likely just because they are extremely big.

Line 62 to 64 and Line 339 to line 340: I completely disagree with these statements: any mutation without RNA expression is unlikely present on the MHC complex as antigen, and thus highly unlikely affect the response of ICB antibody (yes we can’t exclude the possibility of indirect regulation of the expression of other genes: but this is a rare event). Thus, the statement here is misleading.

Minor point

Line 261, the title is incomplete?

Figure S3b, I guess the only recurrent neoantigen is from BRAF V600E: 70% of the melanoma have this mutation.  

There are many typos: for example, line 167 CTLA4 and line 185 SERPINB3.

Reviewer 4 Report

This is an interesting study analyzing the potential genomic indicators 
contributing to ICB therapy response, although the idea is very good, the lack of clinical data (for example: patients charateristics, efficacy of treatment etc etc) low its importance

Author Response

Point1: This is an interesting study analyzing the potential genomic indicators 
contributing to ICB therapy response, although the idea is very good, the lack of clinical data (for example: patients charateristics, efficacy of treatment etc etc) low its importance

Response 1: Thanks for the positive comments on our work. The detailed clinical information was provided in Table S1 as a supplementary file.

Reviewer 5 Report

Major comment:
1) Could the authors investigate whether their P-TMB scoring system is associated with the immune cytolytic activity in (primary and metastatic) skin melanoma and NSCLC tumors? Supporting evidence comes from recent findings, reporting that the TMB was shown to correlate with CYT-high primary (but not metastatic) skin melanomas (Roufas C, et al. Distinct genomic features across cytolytic subgroups in skin melanoma. Cancer Immunol Immunother. 2021;70(11):3137-3154. doi: 10.1007/s00262-021-02918-3). In addition, high CYT levels were reported in lung cancers (Roufas C, et al. The Expression and Prognostic Impact of Immune Cytolytic Activity-Related Markers in Human Malignancies: A Comprehensive Meta-analysis. Front Oncol. 2018;8:27. doi: 10.3389/fonc.2018.00027). 

Minor comments:
1) line 34: it shoud be "INF-γ"
2) line 56: it should be "...thus affecting antitumor response".
3) In Fig. 2C, please indicate which datasets correspond to melanoma (the first 4 appearing in order from left to right). This Figure shows that TMB is higher in 2 out of the 4 NSCLC studies, therefore, the sentence "TMB was significantly higher in responders of melanoma treated by anti-CTLA4 and NSCLC samples" needs to be revised.
4) I think it is better to mention the "impact of paired gene mutations", instead of "mutation impact of a gene pair" (line 228)
5) Please fix "E" -> (e) on the legend of Fig. 3.
6) Line 271: please revise as "... were observed to have significantly higher enrichment score in responders ..."
7) line 274: please revise as "... thus its gene mutation may associate with ..."
8) line 326: please revise as "...may not be applicable".
9) Figure S3 seems to have 2 titles and one legend. Please delete the first ("Figure S3: Recurrent HLAs") and keep the second.
10) Supporting Tables S1 and S2 cannot be found in the manuscript.

Round 2

Reviewer 2 Report

I saw the revision f this manuscript and the authors have indeed
incorporated most of the issues raised. With this improved version of This
manuscript, I reckon that the paper is suitable for publication in "Cells".

Reviewer 3 Report

na

Reviewer 4 Report

In my vision there is no improvement in this paper